ecology/environmental science

bull shark, shark attack, shark–human interactions, tiger shark, white shark

**Author for correspondence:**
Corey J. A. Bradshaw
e-mail: corey.bradshaw@flinders.edu.au

# Predicting potential future reduction in shark bites on people

Corey J. A. Bradshaw[1], Phoebe Meagher[3],
Madeline J. Thiele[1,2], Robert G. Harcourt[4]
and Charlie Huveneers[2]

[1]Global Ecology, College of Science and Engineering, and [2]Southern Shark Ecology Group, College of Science and Engineering, Flinders University, GPO Box 2100, Adelaide, South Australia 5001, Australia
[3]Taronga Conservation Society Australia, Taronga Zoo, Sydney, New South Wales, Australia
[4]Department of Biological Sciences, Macquarie University, Sydney, New South Wales 2109, Australia

CJAB, 0000-0002-5328-7741; RGH, 0000-0003-4666-2934; CH, 0000-0001-8937-1358

Despite the low chance of a person being bitten by a shark, there are serious associated costs. Electronic deterrents are currently the only types of personal deterrent with empirical evidence of a substantial reduction in the probability of being bitten by a shark. We aimed to predict the number of people who could potentially avoid being bitten by sharks in Australia if they wear personal electronic deterrents. We used the Australian Shark Attack File from 1900 to 2020 to develop sinusoidal time-series models of *per capita* incidents, and then stochastically projected these to 2066. We predicted that up to 1063 people (range: 185–2118) could potentially avoid being bitten across Australia by 2066 if all people used the devices. Avoiding death and injury of people over the next half-century is of course highly desirable, especially when considering the additional costs associated with the loss of recreational, commercial and tourism revenue potentially in the tens to hundreds of millions of dollars following clusters of shark-bite events.

## 1. Introduction

Despite the low probabilities of being bitten by a shark (generally less than 1 attack million people$^{-1}$ year$^{-1}$) even in the highest-incident regions of the world [1–4], these events attract disproportionately high attention from the media, the public, politicians and bureaucrats [5,6]. This occurs even with the knowledge that about 15% of all incidents worldwide result in a

human fatality [4], although it is considerably higher (46%) in Réunion [7]. While the disconnect has interesting psychological origins [8], the repercussions for the conservation of shark species are potentially serious. Globally, many mitigation measures are used, including lethal control measures like large-mesh nets and drum lines [9]. While the goal of lethal shark mitigation programmes is to reduce local populations of potentially dangerous sharks, some species targeted by lethal programmes or caught as bycatch are either threatened (e.g. white sharks *Carcharodon carcharias*; green sawfish *Pristis zijsron*—listed as Vulnerable and Critically Endangered, respectively) [10,11], or of conservation concern (e.g. tiger sharks *Galeocerdo cuvier*—74% decline over the past 25 years in eastern Australia) [12–14]. Non-target species are also caught as bycatch (e.g. turtles and dolphins), some of which are legally protected in the same waters [15,16]. The use of such lethal methods can therefore have negative impacts on both target and bycaught species, meaning that such programmes have ecological costs by potentially inhibiting the recovery of threatened species or contributing to further population declines. Other forms of mitigation that are non-lethal include means of increasing the detection of sharks at beaches—for example, aerial spotting from manned planes or unmanned drones [17], or detection of acoustically tagged sharks [18]. In addition, the latest wave of mitigation measures includes a host of personal protective equipment and deterrents worn on the body or equipment such as scuba tanks or surfboards [19–22].

The general consensus is that at least since the 1980s, the rate of *per capita* bites has been increasing worldwide [23], driven mainly by trends in the regions with the highest human populations [4]; regions with the fewest bites have remained unchanged in bite rate [4], while some regions have even had a decrease in *per capita* risk [2]. Most shark bites have been recorded in the USA (52%), rising at an average of 1.07 bites year$^{-1}$ since 1982 [24]. The second-highest number has occurred in Australia, with an increasing rate of 0.35 attacks year$^{-1}$ over a similar period driven mainly by trends in the most populous state of New South Wales [24]. Other, high-incident regions include South Africa (stable) [24], Brazil (increasing by 0.07 attacks year$^{-1}$) [25], Réunion (increasing at 0.05 attacks year$^{-1}$) [7] and the Bahamas (stable) [25]. Given ongoing increases in the number of people engaged in in-water activities [26], it is logical to anticipate increases in shark bites over the coming decades.

The conditions leading to a higher probability of being bitten by a shark have been investigated for decades. Of course, relative shark abundance [27] and the number of people in the waters frequented by sharks [24,28] ultimately influence bite probability to some extent, but to an as-yet unquantifiable degree. In addition, habitat modification and destruction, water quality, climate change and anomalous weather patterns, and the distribution and abundance of prey, have all been hypothesized to explain changes in the number of shark bites and shark bites *per capita* [24,27,29,30]. Indeed, there is evidence that bite probability varies according to the time of day [31,32], water depth [31,32], tide height [32,33], swell height [34], the lunar cycle [35,36], the position of sun [37], the season [3,27,32,33], wind direction [32], temperature [3], rainfall [3], the presence of upwelling [34] and the distance to rivers [3]. Bite probability also varies among species—most worldwide result from white, tiger, bull (*Carcharhinus leucas*) and whaler (e.g. *Carcharhinus limbatus*) sharks [24].

That environmental conditions are correlated with shark-bite behaviour leads to the hypothesis that broad-scale climatic conditions influencing oceanographic and weather conditions could potentially explain long-term trends in rates of shark bites. In Australia where shark bites are relatively common, the Australian Shark Attack File [25] has now amassed over 1100 separate incidences of a shark bite on humans since the late eighteenth century. Despite the conclusion that *per capita* rates have been increasing since the 1980s, examining the country's overall *per capita* bite rate (see Results) over a longer period suggests a potential periodicity in the peaks and lows in bite rate. We hypothesize that this multidecadal variation could potentially be explained in part by long-term patterns of climate variability known as Pacific decadal oscillation. This oscillation is a long-term El Niño-like pattern of Pacific climate variability [38], with 20- to 30-year extremes characterized by widespread variation in the Pacific Basin and the North American climate [39]. For the Australian region, this is potentially best described by the multidecadal El Niño-Southern Oscillation (ENSO) pattern of warm and cool phases [40].

Assuming we can attribute an influence of this type of climate variability on the long-term patterns in bite probability in part to this type of climate variability, it follows that predicting the number of people expected to be bitten by sharks in the future should also be possible given expected human population increases. Of course, this approach relies on many assumptions, the principal being stability in the relative abundance of sharks, shark behaviour, shark distribution (potentially influenced by climate) and human swimming/surfing behaviour and patterns. Without information contravening such assumptions (but see our consideration of relative shark abundance in the Methods), this approach therefore permits assessments of the number of people who could potentially avoid injury or death

from shark bite if they chose to wear appropriately donned electronic deterrents. So far, some personal shark deterrents using an electric field to repel individual sharks have demonstrated measurable and largely consistent success in reducing bite probability [19,21,41–45] by approximately 60% from *Ocean Guardian* electronic-deterrent products in particular.

In this paper, we use the above information to predict the number of ocean/river users who could potentially avoid injury or death from a shark bite by wearing an electronic deterrent operating at optimum capacity. We use the Australian Shark Attack File to demonstrate that national and state-level shark-bite rates *per capita* follow multidecadal patterns of periodicity that in some measure match Pacific decadal oscillation. Using sinusoidal models to fit the patterns combined with projections of human population increase, we simulate the expected number of bites with incrementing proportions of water users employing personal electronic deterrents. We show that despite the low probability of a person being bitten by a shark, appropriate use of personal electronic deterrents could potentially avert over 3000 incidents across Australia by the year 2066.

# 2. Data

## 2.1. Australian Shark Attack File

The Australian Shark Attack File is held at Taronga Zoo (Taronga Conservation Society Australia), Sydney, and is affiliated with the International Shark Attack File. The Australian Shark Attack File is the most comprehensive database of shark–human interactions recorded within Australia's Economic Exclusion Zone. At the time of publication, the database had more than 1100 reported incidents from 1791 through to 2020. The database was founded by John West in 1984, who retrospectively researched incidents prior to this date from a range of sources [46–51] as well as archival print media and internet news service, State and National Library searches, official investigation reports, coronial and police reports, surf life-saving report logs and personal communication.

The File is a dynamic database that is continually researched and is subject to change as new incidents occur or new information becomes available on previously recorded incidents. The database relies on receiving accurate and validated information from scientists and managers in all Australian state fisheries departments. Where possible, the curator will request either the victim, a witness or the state fishery department to complete a File questionnaire (via e-mail). In the cases where bite marks can be analysed by state fisheries departments, the shark species can be further validated forensically. There are 101 fields of information for each incident, including victim details, environmental conditions and details of the shark and injury type.

A 'shark attack' is defined in the database as any shark–human interaction where either a shark (not in captivity) makes a determined attempt to bite a person who is alive and in the water, or the shark bites equipment held by the victim or bites a small-water craft containing the victim. Incidents are catalogued as *unprovoked* or *provoked*. A provoked encounter between a human and a shark is defined as an incident where that person was fishing for, spearing, stabbing, feeding, netting or handling a shark, or where the shark was attracted to the victim by activities such as fishing, spear-fishing or cleaning of captured fish. An unprovoked encounter is when the incident happens without the person being engaged in any of these activities.

While the Australian Shark Attack File is the most comprehensive dataset available for shark–human interactions in Australia, there are limitations to the dataset. First, we could not quantify differences in reporting over time. Advances in technology and communication such as smartphones, means reporting is possibly more likely now for less-severe incidents. Second, differences in state fisheries operations means some departments have more resources to dedicate to validating shark species forensically. This could bias the species represented. Finally, data cannot be confirmed outside of media reports for some incidents if the victim cannot be contacted. Despite these limitations, the dataset is validated against the International Shark Attack File annually and is a valuable resource to investigate patterns and trends in shark-related incidents in Australia.

## 2.2. Human population data

We obtained yearly estimates of the Australian population by state and territory from 1900 to the present, as well as state/territory-specific projections of total population size to 2066, from the Australian Bureau of Statistics (abs.gov.au). We used these data to convert all bite data to *per capita* values by region.

We used the projected population sizes to 2066 to infer the change in numbers of expected victims to 2066 (extent of available population-projection data).

# 3. Methods

## 3.1. Time-series analysis

We examined the raw bite rates (bites year$^{-1}$) for each state and territory from 1900 onwards, and then expressed them as bites person$^{-1}$ year$^{-1}$ across all years and regions based on the population data (described above). We arbitrarily chose 1900 as the start of the series because visual examination suggested data prior to this year would be insufficient to fit statistical models. We scaled all bites person$^{-1}$ year$^{-1}$ data by region (i.e. dividing the data by root mean square) using the *scale* function in R [52].

Our first aim was to examine underlying trends in the long time series going back to 1900. We first calculated temporal autocorrelation with the *acf* and *pacf* functions (option 'demean' set to 'true' in *acf* to use the covariances about the sample means) in R [52], and then estimated a spectral density by a smoothed periodogram based on a fast Fourier transform with the *spec.pgram* function in R, to the scaled bites person$^{-1}$ year$^{-1}$ series.

Examining the long-term trends in scaled bites person$^{-1}$ year$^{-1}$, there was an apparent periodicity operating at a wavelength of approximately 100 years (see Results). To test this, we applied four simplified models to the time series for all of Australia and each region: (i) an intercept-only model ($y \sim x$; no trend), (ii) a linear trend ($y \sim \alpha + \beta x$; decline or increase), (iii) a quadratic relationship ($y \sim \alpha + \beta x + \gamma x^2$; nonlinear) and (iv) a cubic fit ($y \sim \alpha + \beta x + \gamma x^2 + \delta x^3$; short-term periodicity). We implemented each model using the *glm* function in R, setting the error distribution family to Gamma with an identity link. We assessed relative model performance using sample-corrected Akaike's information criterion weights [53] and determined model goodness-of-fit with the percentage of deviance explained.

The Pacific decadal oscillation is a long-term El Niño-like pattern of Pacific climate variability [38], with 20- to 30-year extremes characterized by widespread variation in the Pacific Basin and the North American climate [39]. For the Australian region, this is best described by the long-term ENSO pattern of warm and cool phases, even though ENSO periodicity is typically over much shorter time scales (2–5 years). Nonetheless, the Pacific decadal oscillation index (National Centers for Environmental Information, ncdc.noaa.gov/teleconnections/pdo) is negatively correlated with the ENSO southern oscillation index (Australian Bureau of Meteorology, bom.gov.au/climate/current/soihtm1.shtml—see electronic supplementary material, Appendix S1 and figure S1), with evidence that ENSO also expresses multidecal periodicity over the multi-century scale [40].

To examine the correspondence between the patterns of multidecadal periodicity in both the ENSO (southern oscillation index) and the scaled bites person$^{-1}$ year$^{-1}$ series, we computed a wavelet correspondence between the different series using the *wtc* function in the `biwavelet` package in R [54], setting the mother wavelet function to 'morlet'. We computed a 5-year running mean to the yearly averaged southern-oscillation index to remove the finer-scale periods and focus instead on the longer-term patterns.

## 3.2. Projecting future bites

Based on the hypothesis that the scaled bites person$^{-1}$ year$^{-1}$ series followed a multidecadal periodicity possibly related to broad-scale oceanographic fluctuations, we fitted a sinusoidal function to each region's time series from 2020 to 2066 (the period matching the available population data). This allowed us to calculate the total number of people likely to be bitten over the coming decades (see below). We were obliged to fit a simple sinusoidal function of the form $y \sim \alpha + \beta \text{cosine}(\gamma x + \delta)$ instead of merely extending the cubic or quadratic relationships because the latter would have continued the trends upwards exponentially to unrealistically high values. In fact, the sinusoid function fits the observed model extremely well (see Results).

The fitted sinusoid only projects a median expectation, so to encapsulate the full uncertainty in the projected bites person$^{-1}$ year$^{-1}$ series, we stochastically resampled both the negative and positive residuals of the observed values from the sinusoidal fit (1900–2020). However, the distributions of the negative and positive residuals were non-Gaussian, so we fit separate probability density functions to the histograms using the *density* function in R, selecting the bandwidth using the Sheather & Jones [55] method based on the pilot estimation of derivatives implemented in the function *bw.SJ*. We then

stochastically resampled (with replacement and weighted according to the density functions computed above) both the negative and positive residuals in random order (binomial function with $p = 0.5$), adding these to the median sinusoidal prediction to generate 10 000 future time series of scaled bites person$^{-1}$ year$^{-1}$. Also implemented in the stochastic resampling were two functions (one each to the positive and negative residuals, respectively) limiting the size of the residuals according to a linear relationship between residual size and the period of the sinusoidal function. This reduced the average size of residuals in the sinusoid troughs as in the observed series and tended to increase residuals during sinusoid peaks. This provided a more realistic expression of the temporal trends in variance.

For each region and subset of the data (see below), we then back-transformed the scaled bites person$^{-1}$ year$^{-1}$ projections and multiplied these by the relevant yearly human population projections to provide the total number of people projected to be bitten. We subsetted the data by region (all Australia, New South Wales, Queensland, Western Australia) using all the data, as well as subsetting by bites resulting in fatalities only, unprovoked attacks only, the most commonly reported species (white shark) only, and from the three most common species only: white, tiger and bull sharks. We did not project the data for other states and territories because the lack of data would have made the statistical model fits dubious.

## 3.3. Predicting bites avoided by wearing electronic deterrents

Our main aim was to estimate the number of people that potentially could avoid being bitten by sharks if they chose to wear appropriate electronic deterrents. Of course, to calculate such a potential reduction, it is first necessary to quantify the change in bite probability when wearing a personal deterrent device. In this particular exercise, we do not consider other forms of potentially effective mitigation measures, such as netting, drum lines, drone/aircraft surveillance or other warning devices. Recent evidence suggests that shark deterrents using electric fields have a quantifiable reduction in the probability of a shark bite. Most recently, *Ocean Guardian* products like the *Freedom + Surf* electronic deterrent was shown to reduce the probability of a white shark bite by 56% [19]. We therefore stochastically resampled the back-transformed bites person$^{-1}$ year$^{-1}$ projections and multiplied these by a beta-resampled (assuming 5% standard deviation) reduction probability centred on 0.60, multiplying this product by incrementing proportions of people wearing the devices (from 0.1 to 0.9). This provides a three-dimensional matrix of the reduced number of people expected to be bitten over time (from 2020 to 2066) and by proportion wearing the device. Subtracting this matrix from the equivalent matrix without the reduction gives the matrix expressing the number of people avoiding being bitten by sharks per year to 2066.

## 3.4. Under-reporting sensitivity analysis

Projecting the data from the Australian Shark Attack File in this way assumes a consistent reporting rate through time; however, there is no way to test this assumption given a complete lack of data. Instead, we can examine the sensitivity of the bites averted predicted with our model by assuming a historical sighting bias in the raw data. We therefore generated a completely hypothetical correction series to the underlying data and then reprojected our models. In this scenario, we arbitrarily assumed that at the turn of the twentieth century, only an average of 60% of all shark-bite incidents were recorded. Following a sigmoidal function of the form $y = \dfrac{1 - \gamma}{1 + e^{-\alpha - \beta \log x}}$ (where $y$ = proportion of shark-bite incidents recorded, $\gamma = 0.602$, $\alpha = -1601.064$, $\beta = 211.348$, $x$ = calendar year), we assumed that it would gradually increase during the 1950s and 1960s with the mass uptake of television and then asymptote to 1.0 (full reporting) by the 1990s with the spread of the internet (see electronic supplementary material, Appendix S1 and figure S2a). After applying this correction to the scaled bites person$^{-1}$ year$^{-1}$ series for all of Australia combined (see electronic supplementary material, Appendix S1 and figure S2b), we reprojected the sinusoid model to calculate the total number of bites averted under this under-reporting scenario.

We also considered a second, feasible under-reporting scenario where we set the reporting average at the beginning of the twentieth century to 40% (cf. 60% above), rising again during the television age as above, but asymptoting instead during the 2000s with the rise of social media (where for this model, $\gamma = 0.394$, $\alpha = -1381.997$, $\beta = 182.430$) (see electronic supplementary material, Appendix S1 and figure S3).

## 3.5. Scenarios of relative shark abundance

We also considered two additional scenarios where the probability of being bitten was a function of relative shark abundance over time, based on evidence from Brazil where the rate of shark bites ($r_b$) doubled with a

fivefold increase in the relative abundance ($N_{rel}$) of 'dangerous' sharks [27]; this equates to a power-law relationship of the form: $r_b = N_{rel}^{0.4307}$. Given the evidence for declines in the relative abundance of some of the most commonly attributed shark species to bites in Australia (white, tiger, bull and whaler sharks; see Results) since the early last century [44,56], we constructed a plausible timeline of changing relative abundance for white, tiger and bull/whaler sharks for the states with the most complete data (New South Wales, Queensland and Western Australia) (see full justification for each scenario in electronic supplementary material, Appendix S2). We treated bull and whaler sharks in the same group because many bites attributed to whalers are probably committed by bull sharks [25], and because whaler species have been reported as a group in shark-control programs until the mid- to late 1990s [56,57].

For each scenario, we first examined the relationship between scaled bites person$^{-1}$ year$^{-1}$ species$^{-1}$ and both the southern-oscillation index and the species-specific reconstructions of $N_{rel}$ (with maxima scaled to 1) for these three states separately. We applied simple generalized linear models to the individual state-species time series based on a gamma error distribution and log link function (*glm* R function), fully aware that parameter estimates were likely biased due to temporal autocorrelation (see electronic supplementary material, Appendix S2). However, contrasting the top-ranking models to a comparable generalized least-squares model with an autocorrelation structure based on the *gls* function in the `nlme` R library [58] revealed more information-theoretic support for the generalized linear models in most cases (see electronic supplementary material, Appendix S2). Next, we used the linear forms of the coefficients derived from the bivariate generalized linear model: scaled bites person$^{-1}$ year$^{-1}$ species$^{-1}$ $\sim N_{rel}$ to adjust the observed bite series for each species in each state for both scenarios. Adding the three series together (white, tiger, bull/whaler) sharks, we transformed the annual sums to a proportion of the maximum, and then multiplied the ensuing proportional time series to the total scaled bites person$^{-1}$ year$^{-1}$ for each state. Finally, we repeated all subsequent analyses as described previously to predict new values of averted bites from 2020 to 2066.

## 4. Results

### 4.1. Database summary

From 1900 to the start of 2020, there were 985 incidents reported in the Australian Shark Attack File, from 20 different species reported. Ninety incidents could not be reliably attributed to a single species. Recorded fatalities made up 20% ($n = 194$) of the reported incidents, and 65% ($n = 645$) of all recorded incidents were 'unprovoked'. The most commonly reported species responsible for attacks was the white shark ($n = 309$; 31%), followed by tiger ($n = 203$; 21%), bull ($n = 137$; 14%) and whaler ($n = 62$; 6%; bronze whaler and dusky) sharks, although it is possible that many of the bites attributed to bronze whalers or dusky sharks were in fact committed by bull sharks [25]. Incidents were recorded from nearly all coastal regions of Australia (figure 1), with some notable spatial gaps on the west coast of Tasmania, the Great Australian Bight, the northern-most region of Western Australia, the eastern coast of the Gulf of Carpentaria and the Gippsland coast of Victoria (figure 1).

### 4.2. Time-series analysis

While there was no clear periodicity based on the raw bite rates expressed *per capita* revealed from the smoothed periodogram, there was a positive autocorrelation in the data up to 15–18 years across all data (electronic supplementary material, Appendix S3 and figure S7a); however, partial autocorrelation showed that although there were autocorrelations between subsequent lags, there were few partial autocorrelations apart from 12 years between the first observation and other lags that were not its immediate predecessor (electronic supplementary material, Appendix S3 and figure S7b). The conclusions remain similar when categorized by state (electronic supplementary material, Appendix S3 and figure S7c–h): New South Wales appears to have an autocorrelation at 2, 5 and 17 years (electronic supplementary material, Appendix S3 and figure S7d), Queensland at 2 years (electronic supplementary material, Appendix S3 and figure S7f) and Western Australia at 82 years (electronic supplementary material, Appendix S3 and figure S7 h).

The wavelet correspondence identified several periods of correspondence for the different scaled bites person$^{-1}$ year$^{-1}$ series against the 5-year running mean of average annual southern-oscillation index since 1990, although they were all somewhat out-of-phase. Across all of Australia, there were peaks in correspondence 4–7 years in the 1960s, 1–5 years in the 2000s, 9–16 years and 20–30 years across most

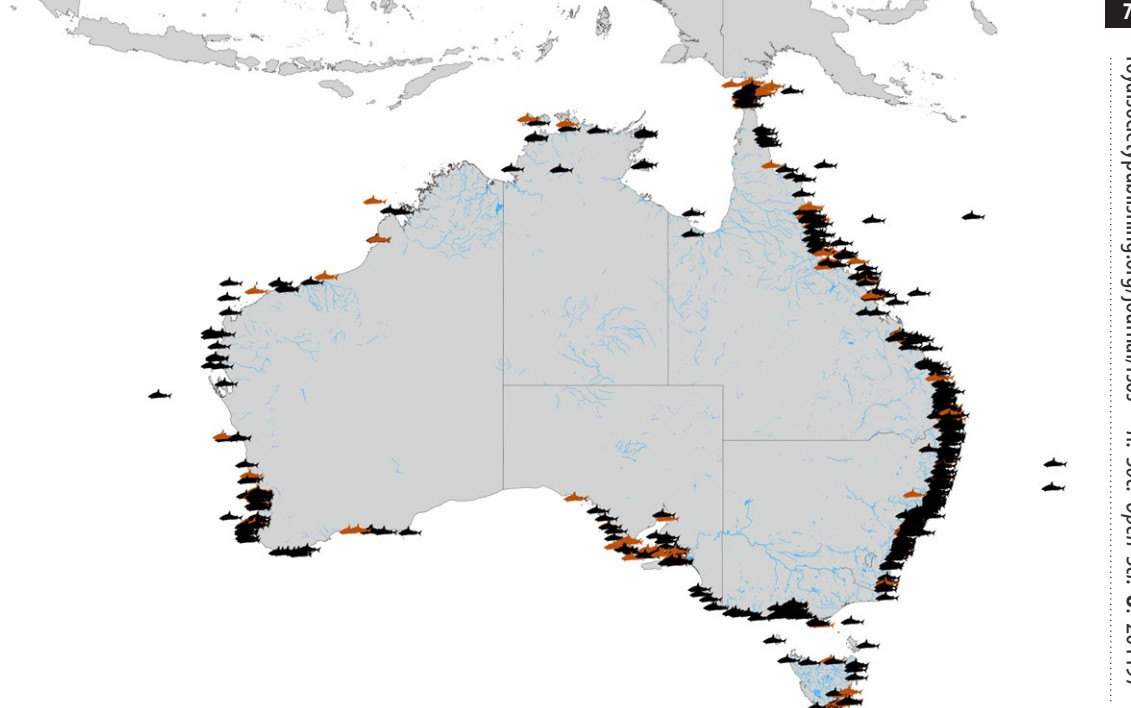

**Figure 1.** Spatial distribution of shark-bite incidents in the Australian Shark Attack File. Red icons show incidents resulting in a human fatality.

of the period since 1900 (figure 2). For New South Wales, 10- to 18-year periodicities showed higher correspondence in the first part of the twentieth century, and were longer (32+ years) over the entire series (figure 2). Queensland and Western Australia had many corresponding cycles below 8 years, with Queensland demonstrating a correspondence at periods of 9–30 years over most of the twentieth century, and Western Australia at periods greater than 30 years throughout the latter half of the twentieth century and into the twenty-first century (figure 2).

### 4.3. Time-series models

For Australia combined, New South Wales and Queensland, the cubic model fitted to the scaled bites person$^{-1}$ year$^{-1}$ series was top-ranked (table 1 and figure 3); for Western Australia, the quadratic model was slightly better supported (table 1; figure 3). We also computed fits to the states and territories with the fewest number of incidents (South Australia, Victoria, Tasmania and Northern Territory (see electronic supplementary material, Appendix S1 and table S1). In all but the Victoria and Tasmania regional subsets, some form of periodicity was supported. Results for the other states are shown in electronic supplementary material figure S8 (electronic supplementary material, Appendix S3).

### 4.4. Future bites averted

Projecting the number of bites person$^{-1}$ year$^{-1}$ to 2066 based on region-specific population projections (figure 3 and electronic supplementary material, Appendix S1 and figure S3) is clearly associated with high uncertainty because of the wide range of residuals from the historical data. However, assuming the underlying sinusoidal pattern is realistic (figure 3), across Australia we could expect a mean of up to 24 people year$^{-1}$ (figure 4) (ranging from less than 10 to more than 50 year$^{-1}$; figure 5) who could avoid being bitten by a shark if everyone engaging in water recreation chose to wear a suitably donned electronic deterrent (figure 4). This translates to a maximum mean of 1063 (95% confidence interval: 185–2121) people averting a shark bite between 2020 and 2066 if all ocean users employed electronic deterrents. Of course, variable uptake of electronic deterrents means that bite avoidance would change accordingly (figure 4); for example, if only half of the ocean users employed electronic deterrents, the number of people averting shark bite would range from 93 to 1061 by 2066. The pattern of predicted bites avoided varies by region following region-specific future sinusoid

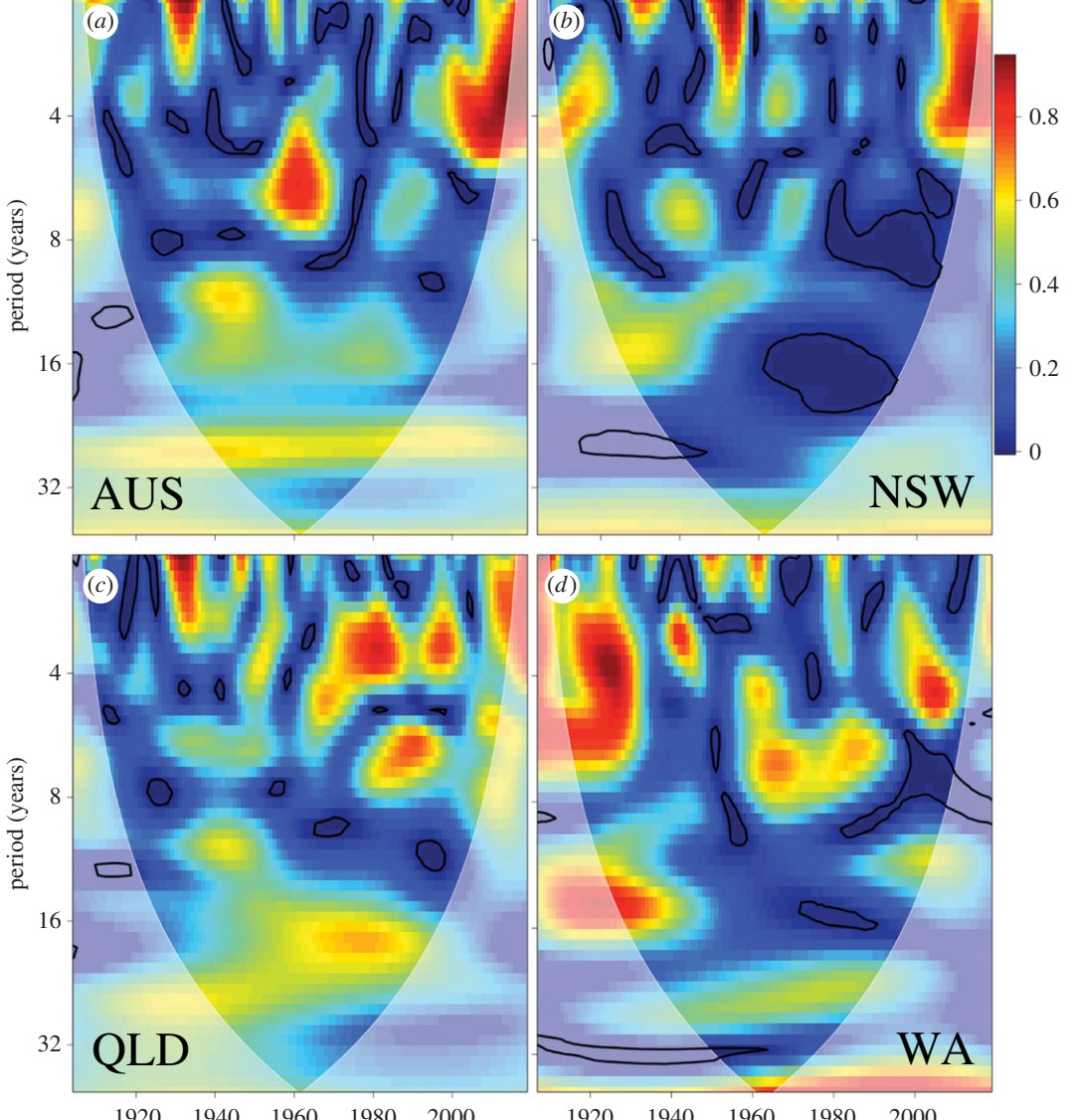

**Figure 2.** Wavelet correspondence between the patterns of multidecadal periodicity in the ENSO (southern oscillation index) and the scaled bites person$^{-1}$ year$^{-1}$ series for (a) Australia combined (AUS), (b) New South Wales (NSW), (c) Queensland (QLD) and (d) Western Australia (WA). The wavelets are computed based on a 5-year running mean of the yearly averaged southern-oscillation index to remove the finer scale periodicity.

projections (figure 4). For the three states with the most complete data, the predicted ranges of averted bites to 2066 were 52–846 (all users wearing) and 26–422 (50% wearing) for New South Wales, 6–1545 (all wearing) and 3–774 (50% wearing) for Queensland, and 300–1045 (all wearing) and 150–523 (50% wearing) for Western Australia.

In the two scenarios where we corrected the scaled bites person-1 year-1 for relative shark abundance (including two forward-projection scenarios), there were some small differences in the predicted total number of bites averted between 2020 and 2066 compared to the uncorrected series (Figure 3 with figures S4 and S5 in electronic supplementary material, Appendix S2). However, the sinusoidal fits were similar in all cases, but with the net effect of reducing the variance in the total predicted number of people averting shark bites for 2020–2066 when using the corrected scenarios (electronic supplementary material, figure S6).

Focussing on the whole of Australia, if we restrict the models to different subsets of the entire Australian Shark Attack File, median maximum predictions of fatalities year$^{-1}$ averted by 2066 would be up to 12, or a total of 324 (28–l004) people between 2020 and 2066 (electronic supplementary material, Appendix S3 and figure S9). For unprovoked attacks, the number averted could exceed 14 year$^{-1}$ (total of 66–l315 people between 2020 and 2066); for white sharks only, more than 4

**Table 1.** Model ranking of the cubic ($\sim$yr + yr$^2$+yr$^3$), quadratic ($\sim$yr + yr$^2$), linear ($\sim$yr) and intercept-only (approx. 1) fits to the scaled bites person$^{-1}$ year$^{-1}$ series for Australia combined, New South Wales (NSW), Queensland (QLD), and Western Australia (WA). For each model, we present the log-likelihood (LL), number of estimated parameters (k), the difference in Akaike's information criterion between the current and top-ranked model ($\Delta$AIC$_c$), the AIC$_c$ weight (wAIC$_c$$\sim$model probability) and the percentage of deviance explained (%DE) as an index of goodness-of-fit. The top-ranked model in each case is shown in italics.

| model | LL | k | $\Delta$AIC$_c$ | wAIC$_c$ | %DE |
|---|---|---|---|---|---|
| *Australia* | | | | | |
| *$\sim$yr + yr$^2$+yr$^3$* | *−40.434* | *4* | *0.000* | *>0.999* | *41.5* |
| $\sim$yr + yr$^2$ | −53.519 | 3 | 23.988 | <0.001 | 27.7 |
| $\sim$yr | −67.830 | 2 | 50.467 | <0.001 | 9.0 |
| $\sim$1 | −73.750 | 1 | 60.201 | <0.001 | — |
| *NSW* | | | | | |
| *$\sim$yr + yr$^2$+yr$^3$* | *−29.631* | *4* | *0.000* | *>0.999* | *50.3* |
| $\sim$yr + yr$^2$ | −47.186 | 3 | 32.887 | <0.001 | 30.2 |
| $\sim$yr | −63.960 | 2 | 64.262 | <0.001 | 3.9 |
| $\sim$1 | −66.065 | 1 | 66.341 | <0.001 | — |
| *QLD* | | | | | |
| *$\sim$yr + yr$^2$+yr$^3$* | *−15.144* | *4* | *0.000* | *0.622* | *59.5* |
| $\sim$yr + yr$^2$ | −16.745 | 3 | 0.998 | 0.378 | 58.4 |
| $\sim$yr | −29.617 | 2 | 24.581 | <0.001 | 47.7 |
| $\sim$1 | −67.332 | 1 | 97.891 | <0.001 | — |
| *WA* | | | | | |
| *$\sim$yr + yr$^2$* | *−7.798* | *3* | *0.000* | *0.648* | *59.1* |
| $\sim$yr + yr$^2$+yr$^3$ | −7.245 | 4 | 1.221 | 0.352 | 59.7 |
| $\sim$yr | −31.292 | 2 | 44.731 | <0.001 | 21.4 |
| $\sim$1 | −40.167 | 1 | 60.294 | <0.001 | — |

incidents year$^{-1}$ (total of 18–603 people between 2020 and 2066) could be averted, and more than 22 year$^{-1}$ for white, tiger and bull sharks combined (total of 175–1893 people between 2020 and 2066; electronic supplementary material, Appendix S3 and figure S9). Assuming an historical under-reporting bias as hypothesized in electronic supplementary material figure S2 (electronic supplementary material, Appendix S1), the maximum number of shark-bite incidents that could be averted increases to 1575 (399–3109) people between 2020 and 2066 (electronic supplementary material, Appendix S3 and figure S10). The second under-reporting scenario hypothesized in electronic supplementary material figure S3 (electronic supplementary material, Appendix S1) raises this value even more (average = 1825 people; ranging from 518 to 3523; electronic supplementary material, Appendix S3 and figure S11).

## 5. Discussion

While a mean maximum of 1063 people out of a projected population size of over 49 million people in Australia by 2066 is a miniscule proportion (0.002%) of people potentially able to avert shark bite (even given 30% would be expected to die), these numbers reflect an important reduction in human suffering and death. Both government actions and popular opinion reflect a high value placed on human–shark interactions, and the ramifications from even a single fatality from shark bite are disproportionally high compared to other causes of human mortality [6,59]. There are associated social, health and economic costs, especially when many bites occur in a particular region over short timeframes (e.g. several months). For example, the New South Wales Government invested AU\$16 million in an

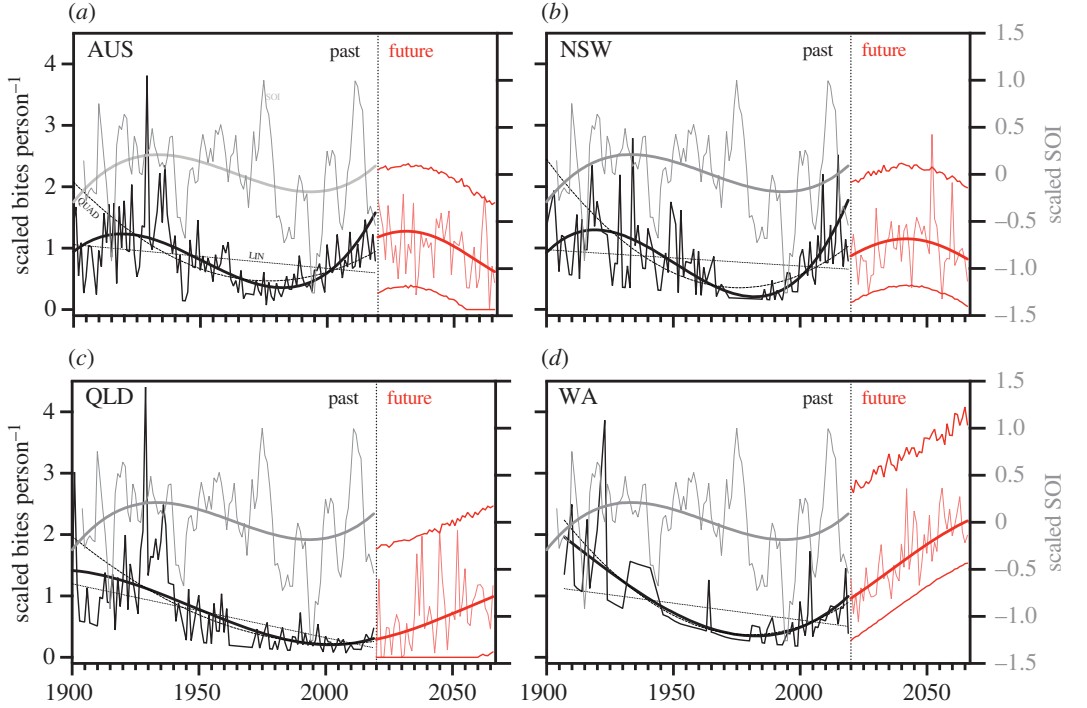

**Figure 3.** Observed scaled bites person$^{-1}$ year$^{-1}$ (black lines) for (*a*) Australia combined (AUS), (*b*) New South Wales (NSW), (*c*) Queensland (QLD) and (*d*) Western Australia (WA). Shown also are the top-ranked modelled fits for the observed series (darker black lines) for each region (table 1). In the (*a*) for Australia, the top-ranked model was the cubic fit (darkest black line), followed by the quadratic (QUAD), and linear (LIN). The yearly averaged southern oscillation index (SOI) is shown in grey (right *y*-axis) with its top-ranked cubic fit (darker grey line). Future (red) sinusoidal fits are shown for 2020–2066, including the median (widest red line) and upper and lower 95% confidence limits (medium-width red lines), and an example iteration sampling from the observed residuals (light red line) for each region.

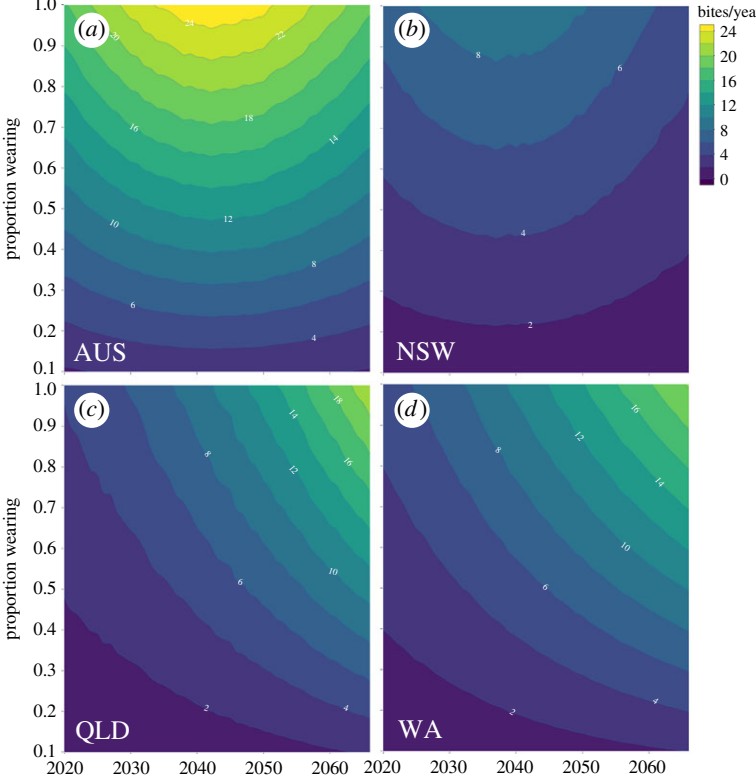

**Figure 4.** Number of fewer bites year$^{-1}$ predicted for (*a*) Australia combined (AUS), (*b*) New South Wales (NSW), (*c*) Queensland (QLD), and (*d*) Western Australia (WA) for incrementing proportions of people wearing electronic deterrents from 2020 to 2066.

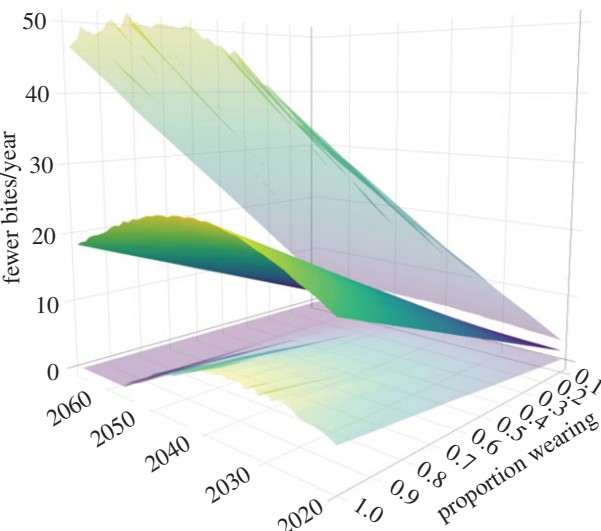

**Figure 5.** Three-dimensional representation of the number of people avoiding shark bites from 2020 to 2066 for incrementing proportions of people wearing electronic deterrents for Australia combined. Shown is the median (darker surface) and the upper and lower 95% confidence limits (lighter surfaces).

attempt to mitigate shark bites [44] in part due to lost revenue from businesses benefitting from in-water users and tourism that could be of the magnitude of tens of millions of dollars [60]. Similarly, surfing was made illegal on Réunion Island (a previously popular area for ocean sport) following a high number of fatal shark bites there, with an estimated cost to the local economy of hundreds of millions of dollars annually [29,61]. Of course, we could never expect 100% uptake, or maximum efficiency, of electronic deterrents, so the actual reduction in injuries and lives lost would likely be somewhat lower. Another potential source of uncertainty is whether sharks might be less inclined to bite surfers and swimmers in groups if only some people in a group are wearing electronic deterrents. However, given that governments are applying multiple approaches to mitigating shark bite (e.g. drones, SMART drumlines, acoustic monitoring), our simulations suggest electronic deterrents could make a valuable contribution to overall mitigation, and so help allay community fears.

Our model predictions are of course subject to many assumptions, most of which are unverifiable. One of the greatest uncertainties here is the relative abundance of the different shark species responsible for most of the incidents. However, our analysis based on reconstructed population trends for the most common species involved in biting people appears to indicate that the predictions are robust to that assumption. Further, if there are large changes in the number of sharks in the vicinity of water goers, then our predictions will likely be biased upward or downward accordingly. However, others have concluded that it is unlikely that the recent trend in shark bite rate is being driven by increasing shark populations [2,25]. Another source of uncertainty comes from potential variation in both the propensity for different shark species to bite people, as well as the temporal and spatial distribution of different size classes of sharks responsible for most bites. Thus, times and regions where the most dangerous individuals within shark populations overlap with people using the water could increase the local incidence of bites beyond what we projected here.

Variability in the distribution of species responsible for bites could also potentially bias our predictions, especially if their main prey sources are responding to climate- or human-driven pressures. Indeed, there is ample evidence now that marine communities on both the west and east coasts of Australia are moving southwards in response to a rapidly warming climate [63–67], including changes in shark densities shifting along the east coast of Australia in particular [68–70]. Other sources of variation are potentially problematic, such as trends in the popularity of ocean/river recreation (bathing, surfing, swimming), trends of human behaviour within these categories, and potentially density-feedback effects (e.g. an increasing likelihood of bites as the density of water goers increases). However, reliable data on the number of people involved in aquatic activities are not available.

Although several studies have demonstrated that electronic deterrents can reduce the probability of shark bites, device efficacy varies among manufacturers [19,43], and depending on the situation tested [41,42]. While some of the differences within and among these studies can be explained by electric field characteristics and the distance between the device and bait/attractant, the efficacy of electronic

deterrents probably depends on the context. We used an approximately 60% reduction in the probability of a bite based on studies showing reductions of 59% [41], 56% [19] and 67% [21]. However, the number of interactions and replicates required in these studies required sharks to be sufficiently motivated to approach the deterrent. Studies are therefore typically done in areas where sharks aggregate and where an attractant such as bait or berley (chum) are applied, making sharks more likely to bite. In addition, shark species, size, or general propensity to bite people (e.g. motivation) could also affect the efficacy of electric deterrents. For example, Colefax *et al.* [62] concluded that most bites occurring on the north coast of New South Wales are by sub-adult white sharks that are most likely not hunting actively. In this situation, electronic deterrents might reduce the likelihood of a bite by more than 60% that we considered here, further increasing the number of lives saved.

# 6. Conclusion

Shark bites are rare events, but they can severely affect victims and their support groups—about one-third of victims report post-traumatic stress disorder after the event [71]. Efforts attempting to reduce the risks of a shark bite, even if the risks themselves are low, are clearly valuable. While estimates of prevented bites we report here are small compared to injuries or death due to many other day-to-day hazards or activities, we should nonetheless attempt to reduce injuries or fatalities from shark bites. Shark bites create not only physical and emotional trauma for the victims, but for water goers in general, leading to a powerful social licence for effective mitigation.

Data accessibility. Data used in this paper are available with permission from Taronga Conservation Society Australia, Taronga Zoo, Sydney. Data and relevant code for this research work are stored in GitHub: https://github.com/cjabradshaw/sharkbite, and have been archived within the Zenodo repository: https://doi.org/10.5281/zenodo.4461747.

Authors' contributions. C.J.A.B. designed the research, did the analyses, and drafted the manuscript. P.M. was responsible for shark-bite data curation and interpretation, and helped draft the manuscript. M.J.T. assessed shark-deterrent efficacy, sourced references and helped draft the manuscript. R.G.H. provided economic and social-science aspects in the text, contributed data to sensitivity analyses and helped draft the manuscript. C.H. helped design the research, compiled the scenario-correction data for shark abundance and helped draft the manuscript. All authors gave final approval for publication.

Competing interests. We declare we have no competing interests.

Funding. Funded by the Flinders University Marine and Coastal Research Consortium (Flinders.edu.au/Marine-Research).

Acknowledgements. We thank C. Simpfendorfer for helpful comments to improve the manuscript.

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
