## [Peer Review File · Royal Society Open Science]

Review History

RSOS-201197.R0 (Original submission)

Review form: Reviewer 1 (Brendan Kelaher)

Is the manuscript scientifically sound in its present form?

Yes

Are the interpretations and conclusions justified by the results?

Yes

Is the language acceptable?

Yes

Do you have any ethical concerns with this paper?

No

Have you any concerns about statistical analyses in this paper?

No

Recommendation?

Accept with minor revision (please list in comments)

Comments to the Author(s)

This manuscript describes how a practical electronic deterrent could potentially reduce shark bite incidents going forward. The manuscript is well-written, and the analysis of data was robust and rigorous. I particularly liked the results about how variation in shark bites could be explained by long-term patterns of climate variability, such as the Pacific decadal oscillation. I have a few minor comments below that should be addressed. Following that, I recommend publication.

Line 31 "potentially thousands" is a little ambiguous.

Line 41 Change the second "the" to "these".

Line 56 This is perhaps a little overstated. The cost is perhaps "high" locally, but in general the mortality of target and bycatch species from shark control methods is substantially less than incidental bycatch from fishing industries.

Line 124 "avert thousands" is a little ambiguous. I think you need to be more precise.

Line 245 "Culprit" is generally reserved for misdeeds or crimes, which is perhaps not appropriate for sharks who are just going about their general business.

Line 257 The modelling is based on a 58% reduction of shark bites from the use of an electronic device for data collected for relatively large white sharks in SA. How well does this translate to other sharks species or white sharks of different sizes? For example, most bites on the NSW north coast are coming from sub-adult white sharks, who are most likely not actively hunting or excited by food see <https://doi.org/10.3389/fmars.2020.00268>. I note this is covered in the third paragraph of the discussion, but perhaps a little more text is needed on variation among species and sizes of sharks.

Line 383 When usage is at 50%, would bite probabilities be impacted by the fact that water users (surfers and swimmers) are often in groups.

Line 419 "Immense" is perhaps a little strong. For example, \$16 million AUD is just the cost of a couple of hundred metres of highway upgrades.

Line 441 Change "culprit".

Decision letter (RSOS-201197.R0)

Dear Professor Bradshaw

On behalf of the Editors, we are pleased to inform you that your Manuscript RSOS-201197 "Predicting potential future reduction in shark bites on people" has been accepted for publication

in Royal Society Open Science subject to minor revision in accordance with the referees' reports. Please find the referees' comments along with any feedback from the Editors below my signature.

Please submit your revised manuscript and required files (see below) no later than 7 days from today's (ie 22-Jan-2021) date. Note: the ScholarOne system will 'lock' if submission of the revision is attempted 7 or more days after the deadline. If you do not think you will be able to meet this deadline please contact the editorial office immediately.

on behalf of Pete Smith (Subject Editor)
openscience@royalsociety.org

Reviewer comments to Author:
Reviewer: 1

Comments to the Author(s)

This manuscript describes how a practical electronic deterrent could potentially reduce shark bite incidents going forward. The manuscript is well-written, and the analysis of data was robust and rigorous. I particularly liked the results about how variation in shark bites could be explained by long-term patterns of climate variability, such as the Pacific decadal oscillation. I have a few minor comments below that should be addressed. Following that, I recommend publication.

Line 31 "potentially thousands" is a little ambiguous.

Line 41 Change the second "the" to "these".

Line 56 This is perhaps a little overstated. The cost is perhaps "high" locally, but in general the mortality of target and bycatch species from shark control methods is substantially less than incidental bycatch from fishing industries.

Line 124 "avert thousands" is a little ambiguous. I think you need to be more precise.

Line 245 “Culprit” is generally reserved for misdeeds or crimes, which is perhaps not appropriate for sharks who are just going about their general business.

Line 257 The modelling is based on a 58% reduction of shark bites from the use of an electronic device for data collected for relatively large white sharks in SA. How well does this translate to other sharks species or white sharks of different sizes? For example, most bites on the NSW north coast are coming from sub-adult white sharks, who are most likely not actively hunting or excited by food see <https://doi.org/10.3389/fmars.2020.00268>. I note this is covered in the third paragraph of the discussion, but perhaps a little more text is needed on variation among species and sizes of sharks.

Line 383 When usage is at 50%, would bite probabilities be impacted by the fact that water users (surfers and swimmers) are often in groups.

Line 419 “Immense” is perhaps a little strong. For example, \$16 million AUD is just the cost of a couple of hundred metres of highway upgrades.

Line 441 Change “culprit”.

===PREPARING YOUR MANUSCRIPT===

===PREPARING YOUR REVISION IN SCHOLARONE===

To revise your manuscript, log into <https://mc.manuscriptcentral.com/rsos> and enter your Author Centre - this may be accessed by clicking on "Author" in the dark toolbar at the top of the

page (just below the journal name). You will find your manuscript listed under "Manuscripts with Decisions". Under "Actions", click on "Create a Revision".

<https://royalsociety.org/journals/authors/author-guidelines/#supplementary-material> to include a suitable title and informative caption. An example of appropriate titling and captioning may be found at https://figshare.com/articles/Table_S2_from_Is_there_a_trade-off_between_peak_performance_and_performance_breadth_across_temperatures_for_aerobic_sc_ope_in_teleost_fishes_/3843624.

Author's Response to Decision Letter for (RSOS-201197.R0)

See Appendix A.

Decision letter (RSOS-201197.R1)

Dear Professor Bradshaw,

It is a pleasure to accept your manuscript entitled "Predicting potential future reduction in shark bites on people" in its current form for publication in Royal Society Open Science.

on behalf of Professor Pete Smith (Subject Editor)
openscience@royalsociety.org

Appendix A

Reviewer: 1

Line 31 “potentially thousands” is a little ambiguous.

RESPONSE: We have just removed this qualifying expression. It now just reads “Avoiding death and injury of people ...”

Line 41 Change the second “the” to “these”.

RESPONSE: Changed.

Line 56 This is perhaps a little overstated. The cost is perhaps “high” locally, but in general the mortality of target and bycatch species from shark control methods is substantially less than incidental bycatch from fishing industries.

RESPONSE: We have now changed this sentence to:

“The use of such lethal methods can therefore have negative impacts on both target and bycaught species, meaning that such programs have ecological costs by potentially inhibiting the recovery of threatened species or contributing to further population declines.”

Line 124 “avert thousands” is a little ambiguous. I think you need to be more precise.

RESPONSE: We have changed this to:

“... could potentially avert over 3000 incidents across Australia ...”

Line 245 “Culprit” is generally reserved for misdeeds or crimes, which is perhaps not appropriate for sharks who are just going about their general business.

RESPONSE: We have removed the word ‘culprit’.

Line 257 The modelling is based on a 58% reduction of shark bites from the use of an electronic device for data collected for relatively large white sharks in SA. How well does this translate to other sharks species or white sharks of different sizes? For example, most bites on the NSW north coast are coming from sub-adult white sharks, who are most likely not actively hunting or excited by food see <https://doi.org/10.3389/fmars.2020.00268>. I note this is covered in the third paragraph of the discussion, but perhaps a little more text is needed on variation among species and sizes of sharks.

RESPONSE: Agreed, but we think this concept is better addressed in the third paragraph as an additional source of uncertainty. We have therefore added the following text to the end of the third paragraph in the Discussion:

“Another source of uncertainty comes from potential variation in the both the propensity for different shark species to bite people, as well as the temporal and spatial distribution of different size classes of sharks responsible for most bites. For example, Colefax et al. [61] concluded that most bites occurring on the north coast of New South Wales are by sub-adult white sharks that are most likely not actively hunting. Thus, times and regions where the most dangerous individuals within shark populations overlap with people using the water could increase the local incidence of bites beyond what we projected here.”

Line 383 When usage is at 50%, would bite probabilities be impacted by the fact that water users (surfers and swimmers) are often in groups.

RESPONSE: Interesting point. Instead of addressing this possibility here, we have added the following sentence to the first paragraph of the Discussion (penultimate sentence):

“Another potential source of uncertainty is whether sharks might be less inclined to bite surfers and swimmers in groups if only some people in a group are wearing electronic deterrents.”

Line 419 “Immense” is perhaps a little strong. For example, \$16 million AUD is just the cost of a couple of hundred metres of highway upgrades.

RESPONSE: We have removed the word ‘immense’.

Line 441 Change “culprit”.

RESPONSE: Now changed to “... species responsible for bites ...”